# HIV-Associated Systemic Sclerosis: Literature Review and a Rare Case Report

**DOI:** 10.3390/ijerph191610066

**Published:** 2022-08-15

**Authors:** Shamimul Hasan, Mohd. Aqil, Rajat Panigrahi

**Affiliations:** 1Department of Oral Medicine and Radiology, Faculty of Dentistry, Jamia Millia Islamia University, New Delhi 110025, India; 2Department of Oral Medicine and Radiology, Institute of Dental Sciences, Siksha ‘O’ Anusandhan University, Bhubaneswar 750017, India

**Keywords:** autoimmune diseases, antiretroviral therapy, highly active, HIV, HIV-associated systemic sclerosis, orofacial features, scleroderma, systemic

## Abstract

Highly antiretroviral therapy (HAART) used in Human Immunodeficiency Virus (HIV) treatment may prolong the life span of people living with HIV/Acquired Immune Deficiency Syndrome (AIDS) but may also induce the onset of autoimmune disorders. However, HIV-associated systemic sclerosis (SSc) is an extremely rare occurrence, and only four case reports and two studies documenting this association have been reported to date. We report a rare case of HIV-associated SSc who was referred to us for pain management in her mandibular teeth. A 44-year-old female patient diagnosed with HIV-associated SSc reported a complaint of pain in the lower posterior teeth region. Physical examination revealed typical features of SSc. The pain in her mandibular teeth was due to food lodgement, and she was advised to use toothpaste with a powered toothbrush and mouth stretching exercises, followed by oral prophylaxis. The patient responded well to therapy. HIV-associated SSc is an extremely rare occurrence, with an obscure pathogenic mechanism of HIV-associated autoimmunity. Oral physicians play a crucial role in disease management and should be incorporated into the multidisciplinary team.

## 1. Introduction

Human immunodeficiency virus (HIV) was initially described in the year 1981. Since then, breakthrough advancements have taken place in the diagnostic and therapeutic protocols, thereby influencing the disease prognosis and prevalence [1]. Over the years, HIV infection has emerged as a universal public health menace. The 2018 statistics of the Joint United Nations Programme on HIV and AIDS (UNAIDS) reported approximately 37.9 million individuals living with HIV, with a 0.8% global HIV infection rate [2].

Highly active antiretroviral therapy (HAART) has a favourable prognosis for people living with HIV/AIDS (PLWHA) by prolonging the life span. However, it may also induce the onset of other co-morbid states, including autoimmune diseases [3].

HAART may act as a double-edged sword. It may diminish the autoimmune episode by restoring the dysregulated immune function [4]. It may also provoke the exacerbation of a latent autoimmune disorder or may trigger a new autoimmune disease’s onset during the immune restoration inflammatory syndrome (IRIS), simulating a delayed hypersensitivity reaction to a foreign or self-antigen [4].

The occurrence of rheumatologic ailments in HIV infection dates way back to the 1980s, and the array of rheumatic conditions (arthralgia, arthritis, myalgia, and myositis) have evolved significantly in the post-HAART period. However, autoimmune connective tissue diseases (such as Sjögren’s syndrome, systemic lupus erythematosus, dermatomyositis, and systemic sclerosis) have infrequently been documented with HIV [5].

The coexistence of autoimmune and inflammatory diseases despite an apparent loss of immunocompetence caused by HIV infection is quite ambiguous [6]. Hence, such cases need to be reported to delineate the complex pathogenic mechanisms involved in HIV-associated autoimmunity. Whether the autoimmune diseases occurred due to HIV itself, HIV-related dysregulated immune system, or molecular simulation between the HIV and self-antigen needs to be further explored.

Systemic sclerosis (SSc) is a rare entity affecting less than 10 per 100,000 individuals annually in both Europe and North America [7]. Patients often complain of dry mouth due to salivary gland fibrosis. Dry-mouth symptoms may predispose to an increased incidence of candida infections and high caries rates [8,9]. Dysphagia and dysphonia may be seen due to tongue stiffness and limited mobility, along with decreased flexibility of the oral mucosa and soft palate [10]. Tooth erosion and dental caries may be seen in cases of patients with oesophagal dysmotility, probably due to the gastroesophageal reflux effect [11]. Fibrous adhesions in the articulating surfaces of the temporomandibular joint may result in pseudoankylosis [9]. Aberrant mandibular movements and subluxation are the other likely dental features. Advanced vascular fibrosis and delayed wound healing pose a challenge to surgical treatment [12]. Periodontal features include a low gingival bleeding index, increased periodontal pocket depth, higher periodontal attachment loss, and multiple foci of gingival recession [13].

Infrequently, SSc may be associated with idiopathic tooth resorption. Extreme resorption may result in painful trigeminal neuropathy (TN) due to compression of the inferior alveolar nerve [14]. These patients are more vulnerable to developing oropharyngeal and tongue carcinoma; hence, these patients should be subjected to a detailed and periodic oral cancer screening, even in cases with a restricted oral opening (e.g., by using an endoscope) [15].

An extensive literature search carried out on the Google Scholar and PubMed search engines revealed only four case reports of HIV-associated SSc [5,16,17,18]. To the best of our knowledge, ours is the fifth reported case. Hereby, we report a rare case of HIV-associated SSc in a 44-year-old female who reported mild pain in the mandibular posterior teeth region and responded well to the prescribed treatment.

## 2. Case Report

A 44-year-old female patient was referred to our outpatient department (on 8 April 2021) for the complaint of mild, dull, intermittent pain, localized to the lower posterior teeth region, for the last 10–15 days. Her medical history reveals that she was diagnosed with HIV infection in 2012 (retro-positive from her husband). The patient is currently on a TLE regimen of antiretrovirals (tenofovir + lamivudine + efavirenz). Gradually, she developed tightening of the skin with reduced manual dexterity and was diagnosed with SSc in 2017. (Her laboratory investigations and treatment regimen are summarized in Table 1.)

There was considerable weight loss with fluctuating jaundice since 2016, and her blood sugar values and blood pressure were within the normal range. Physical examination revealed that the patient was thin built and malnourished, calm, cooperative with normal intelligence, and responding to commands. Extraoral examination revealed smooth, tense, shiny, non-pinchable skin with a loss of wrinkles and wide-open eyes (mask-like facies). Lower palpebral conjunctiva elicited a pallor appearance. The skin over the forehead appeared dry and hyperpigmented with interspersed areas of hypopigmented macules (“salt and pepper” appearance). Atrophic nasal alae contributed to a sharp pinched appearance of the nose (mouse facies). Reduced oral aperture (microstomia) with circumoral radiating furrows (rhagades) and thinned lips resulted in a “purse-string/fish mouth” appearance (Figure 1A–C). The skin over the hands was hard, smooth, shiny, non-pinchable, and exhibited a “salt and pepper” appearance. Stiffened phalangeal joints on the fingers with reduced manual dexterity and mild resorption of the terminal phalanges were also observed (Figure 1D,E).

Intraoral examination revealed limited mouth opening (22 mm) and decreased intercommisural distance (26 mm). Buccal and labial mucosa were pale, blanched, with palpable fibrotic bands. The tongue was depapillated, rigid, and with restricted protrusive movements. A positive mouth mirror test was suggestive of xerostomia. Generalized marginal gingival inflammation, a mild recession in lower anterior teeth, and generalized pockets were also seen, which may be the cause of pain due to food lodgement in the mandibular posterior teeth region (Figure 2).

Orthopantomogram (OPG) showed generalized PDL space widening, especially pronounced in the right and left mandibular and left maxillary posterior teeth, with mild interdental bone loss seen. Mild flattening of the left condyle was also noticed (Figure 3).

The clinical, radiographic, and serological parameters were suggestive of HIV-associated diffuse SSc. The patient was educated and motivated for oral hygiene maintenance. Mouth stretching exercises (ice cream sticks/tongue blades), use of a toothpaste with a powered toothbrush, and antiseptic mouthwash were explained to the patient. The patient was reviewed after a month and reported improvement in mouth opening, and was then subjected to oral prophylaxis. GutCade probiotics were also advised as maintenance therapy for the regulation of eubiosis. The patient is currently under follow-up with an increased mouth opening (30 mm) (Figure 4).

## 3. Discussion

The post-HAART era has witnessed a substantial diminution in the prevalence of rheumatic features with HIV infection. However, the emergence of another class of rheumatic disorders encompassing an array of systemic autoimmune and autoinflammatory disorders has gained immense concern and significance and resulted in a new clinical and therapeutic dilemma [19,20,21,22,23].

HIV-associated SSc is an infrequent occurrence, and only four case reports [5,16,17,18] and two studies [24,25] have delineated the association of HIV infection with SSc (Table 2).

Calabrese et al. (2005) evaluated the association of autoimmune disorders with HIV infection in an array of 32 published cases. Most of the cases exhibited de novo autoimmunity, while about 20% described the exacerbation of preceding disorders. The onset of these autoimmune diseases also exhibited significant variation, ranging from occurring briefly after induction of HAART to 27 months, with a mean of 9 months [20]. Cases reported by Dembele et al. [16] and Sikdar et al. [18] demonstrated the simultaneous occurrence of HIV and SSc. However, our findings were in concurrence with the other case reports, which demonstrated that SSc was diagnosed after some time of evolution of the immunosuppression and/or the treatment of HIV [5,17]. Our patient developed features of SSc after 5 years of antiretroviral therapy, most likely due to immune restoration syndrome.

The precise pathogenesis of HIV-associated autoimmunity is still obscure. However, the role of retroviruses, HIV-evoked immune dysfunction causing an autoimmune state, molecular simulation between the viral protein and self-antigens (molecular mimicry), and autoimmune diseases depicting a type of IRIS are the various proposed etiopathogenic mechanisms. HIV infection and autoimmunity may also exist as separate ailments, and the onset of autoimmune disorders may be seen during HIV-evoked immune deficiency [3].

Few affirmative links have advocated the retroviral role in the etiopathogenesis of autoimmune disorders. However, large cohort studies are needed to establish the precise prevalence of autoimmune diseases with HIV [25]. The demonstration of retroviral conserved pol sequences in the sera of systemic sclerosis and mixed connective tissue disease patients (MCTD) may suggest the possible role of retroviruses in the pathogenesis of autoimmune diseases, thus advocating that these sequences may correspond with the onset of an autoimmune reaction to U1-70 kDa polypeptide [26]. Few studies have also exhibited SSc-like extracellular matrix protein production by normal human skin fibroblasts following retroviral protein expression [27]. Long-standing HIV infection results in a gradual diminution of humoral immunity owing to a consistent reduction in the CD4+ T-cell count. Compensatory stimulation of B-Cell proliferation with the increased production of immunoglobulins (mostly abnormal) occurs, and this dysregulated immune mechanism may increase the likelihood of autoimmune disease development [1]. HIV infection may evoke autoantibody production as there is structural antigen simulation between HIV proteins and self-antigens. This molecular simulation between the viral protein and self-antigens causes the antibodies to cross-react and eventually leads to the onset of autoimmune diseases. HIV infection may also result in T-lymphocytes disparity, typified by decreased CD4+ T cells and a proportionate increase in CD8+ T cells. This HIV-evoked T lymphocyte disparity may also attribute to the onset of autoimmune diseases [25]. Published literature has revealed the sequence congruity between viral proteins and self-antigens [28,29,30] (Table 3).

Recent national recommendations in the USA advocate a combination of two nucleoside reverse transcriptase inhibitors (NRTIs) with an integrase strand transfer inhibitor (INSTI) as initial therapy for ART-naïve individuals [31]. However, lifelong ART therapy is mandatory, with no definitive cure. Once the therapy is discontinued, there is a resurgence of the infection. The literature has reported that a combination of immune activators, therapeutic vaccines, and neutralizing antibodies have provided cures in some nonhuman primates, thus posing a hope for future treatment strategies in humans. In vivo delivery of gene-editing tools to either target the virus, boost immunity, or protect cells from infection may also serve as future HIV treatment strategies [32].

SSc is a multisystemic autoimmune disorder characterized by microvascular and immune abnormalities, resulting in cutaneous and internal organ fibrosis. Thick, hardened skin is the distinctive disease manifestation; hence, it has also been termed a “hidebound disease” [8,33]. The disease is categorized into two major forms—(a) a diffuse form, with extensive skin involvement and rapid advanced internal organ involvement, and (b) a localized form, with limited skin involvement (distal of the elbows, knees, fingers, and face). The extent of disease progression, internal organ involvement, and production of specific antibodies form the differentiation basis of these forms [8].

Environmental factors may also induce the development of autoimmune disorders. Exposure to infections (*H. pylori,* cytomegalovirus, Ebstein–Barr virus), chemicals (silica, organic solvents, xylene), heavy metals (mercury, palladium), and drugs (bleomycin, antiretrovirals) have been proposed as potential triggers for SSc development [34]. The published literature has revealed that scleroderma-like cutaneous lesions have been observed with the injectable fusion inhibitor antiretroviral drug enfuvirtide [35]. Another antiretroviral integrase inhibitor drug, raltegravir, has also been associated with the flares of an autoimmune episode [5].

However, our patient did not report any exposure to these environmental factors, nor was she on any medication known to produce scleroderma-like lesions.

The skin over the extremities, face, and trunk may exhibit hyperpigmentation, with distinct depigmented regions. A “salt and pepper” appearance, characterized by vitiligo-like depigmentation and perifollicular hyperpigmentation, is a distinguishing skin manifestation. This typical appearance, together with skin sclerosis, is a diagnostic sign of SSc [9]. Terminal phalanges resorption, short and claw-like fingers (due to acro-osteolysis), and fingertip ulcerations are also usually seen [8].

Our patient presented a typical “salt and pepper” appearance, predominantly over the forehead and hands. Thick, adherent skin over the extremities, stiffened phalangeal joints on the fingers, reduced dexterity, and mild resorption of terminal phalanges were also observed.

Orofacial manifestations may be seen in approximately 80% of SSc cases. Excessive perioral and periorbital sclerosis result in generalized thickened, non-pinchable skin with smooth, shiny texture, comprising the pathognomic “expressionless,” “mask-like,” or “Mona Lisa” facies or “wide-eyed” appearance [36]. “Mouse facies” result from atrophied nasal alae, giving a sharp-nosed appearance. Lip and perioral sclerosis result in limited mouth opening (microstomia) and width (microcheilia) and distinctive creases radiating from the mouth, giving a “purse-string-like” appearance. Patients often complain of xerostomia, predisposing them to an increased incidence of candidiasis and high caries rates [8,9]. Periodontal features include a low gingival bleeding index, increased periodontal pocket depth, higher periodontal attachment loss, and multiple foci of gingival recession [9,13].

Our patient reported typical orofacial manifestations of SSc—thick, shiny facial skin, expressionless appearance, atrophied nasal alae, reduced interincisal and intercommissural distance with pale, fibrotic mucosa, and a depapillated tongue. Gingival recession in the lower anterior teeth and deep pockets were also seen.

Periodontal ligament space widening and mandibular erosions are the distinctive radiographic manifestations of SSc [8]. Periodontal ligament space widening is the most frequently occurring radiographic presentation, and SSc should be given a place in the differential diagnosis, where the radiographic features demonstrate widened PDL space with intact lamina dura (particularly in non-mobile posterior teeth with widened PDL space in more than one segment) [36].

Another radiographic manifestation reported in these patients is the presence of bilateral, sharply defined, relatively uniform mandibular erosions at the insertion sites of masticatory muscles (angles, coronoid process, condyles, or digastric region) [36]. The blunting of mandibular angles results in a characteristic “tail of the whale” radiographic appearance [9].

The radiographic findings seen in our patient included a widened PDL space (more pronounced in posterior teeth) with intact lamina dura and a slight flattening of the left condylar head.

The treatment of SSc necessitates an integrative strategy by incorporating systemic and dental management. The treatment is aimed at inhibiting the production and the accumulation of fibrous proteins, inhibiting vasculature dysfunction, and diminishing the inflammation. Medical management is aimed at preventing tissue ischemia (calcium channel blockers), inhibiting inflammatory-immune processes (immunosuppressive drugs—methotrexate, cyclophosphamide, mycophenolate mofetil, cyclosporine A), and inhibiting excess collagen synthesis (D-penicillamine, colchicine) [14].

Most SSc patients present gastrointestinal tract dysfunction and dysbiosis (altered gut microbiota composition) [37]. Paraprobiotics and postbiotics play a proactive role in the maintenance of eubiosis. Paraprobiotics are a novel ancillary treatment strategy for periodontal disease. Paraprobiotics not only serve as an efficacious regimen for the domiciliary maintenance of oral health but also assess the cellular and inflammatory variables due to their immunomodulatory effects [38]. Recently, a postbiotic-based gel containing lactoferrin and aloe barbadensis leaf juice powder was used as a treatment regimen for periodontitis [39].

Numerous medications used as disease-modifying agents in SSc may exhibit orofacial adverse effects. Calcium-channel blockers, primarily used to treat Raynaud’s phenomenon and other vascular manifestations, may evoke gum hypertrophy [40]. Maintaining meticulous oral hygiene along with drug substitution may result in remission of drug-induced gum hyperplasia [41]. Oral ulceration can also be a side effect of SSc pharmacological treatment, which includes methotrexate, azathioprine, and cyclophosphamide, thus requiring substitution of the drugs [15]. The severity of Sicca syndrome may be enhanced in patients receiving anticholinergic antidepressants for psychiatric ailments [42]. Jaw osteonecrosis may result in patients receiving bisphosphonates along with prolonged steroid therapy, either to prevent osteoporosis or to treat corticosteroid-induced osteoporosis. Thus, a detailed oral examination encompassing optimal periodontal health, removal of unrestorable teeth, and completion of all invasive dental procedures is essential before initiating bisphosphonate therapy. Anti-vitamin K anticoagulants may result in spontaneous or induced gingival bleeding, particularly in patients with poor oral hygiene. Anticoagulant therapy should be initiated cautiously in patients taking antifungal therapies and necessitates repeated INR measurements [40].

The oral features in systemic sclerosis necessitate an interdisciplinary and preventive approach [40]. Patients should be educated and encouraged to maintain meticulous oral hygiene [43]. The significance of oral hygiene may be emphasized by instructing proper brushing techniques, thus preventing caries and periodontal diseases (gingivitis, periodontitis, gingival hyperplasia, etc.). The oral sessions should be short and carried out early in the day. These patients should be reviewed biannually to identify any pathology [40].

The diminished manual dexterity due to sclerodactyly, together with inaccessibility due to microstomia, hampers toothbrush manoeuvering, making oral hygiene maintenance difficult. Adapted tools, such as powered toothbrushes and floss forks, may be of help to overcome this limitation [34]. Perioral exercise programs should be carried out regularly. Combined treatments, including physical therapy, mouth stretching exercises, or massages, may result in an improved oral opening [14]. Patients should be advised to have regular physical therapies (tongue blade/ice cream stick exercises) and occupational remedies. This will not only ameliorate the mouth opening but also aid in minimizing contractures [33,43]. Microstomia may further hinder impression-taking for prosthetic rehabilitation. Additionally, soft liners should be used at the base of the prostheses to minimize pain and discomfort while using prostheses made of rigid materials [40].

Xerostomia and xerostomia-related complications (dysgeusia, candidiasis, and dental caries) may be prevented by the use of fluoridated toothpaste, sugar-free candies, artificial saliva, and sialagogues [33,43]. Mouthwash containing baking soda and systemic fluoridation with fluorine-carrying devices and fluoride varnishes are helpful.

Oral ulcers can be managed by the topical application of anaesthetics and antiseptics (such as lidocaine 2% and chlorhexidine) [40].

Our patient was advised to practice mouth-stretching exercises (ice cream sticks/tongue blades) and use a toothpaste and a powered toothbrush, followed by oral prophylaxis. The pain has subsided, and the mouth opening has also improved.

## 4. Conclusions

HIV-associated systemic sclerosis is an extremely rare occurrence with an obscure pathogenic mechanism of HIV-associated autoimmunity. Oral physicians play a pivotal role in disease management and should be incorporated into the multidisciplinary team. Periodic follow-up emphasizing the maintenance of meticulous oral hygiene, together with physical therapies, such as using tongue blades/ice cream sticks, are necessary measures to maintain oral health in these patients. Probiotics should be incorporated as maintenance therapy for the regulation of eubiosis.

## Figures and Tables

**Figure 1 ijerph-19-10066-f001:**
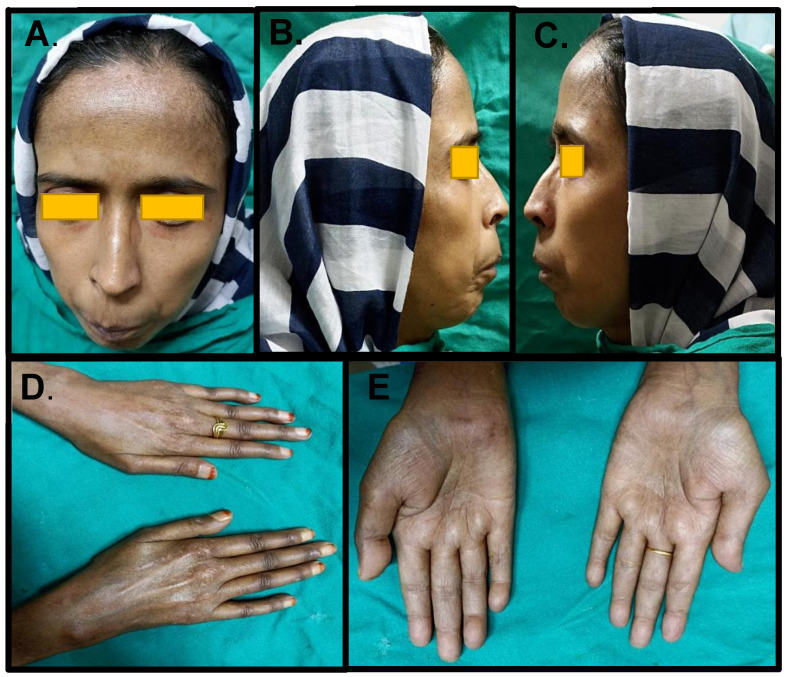
(**A**)—Mask-like facies and salt and pepper appearance of the forehead (**B**,**C**)—pinched nose, thin and incompetent lips, and (**D**,**E**)—thick skin over hands and stiff joints.

**Figure 2 ijerph-19-10066-f002:**
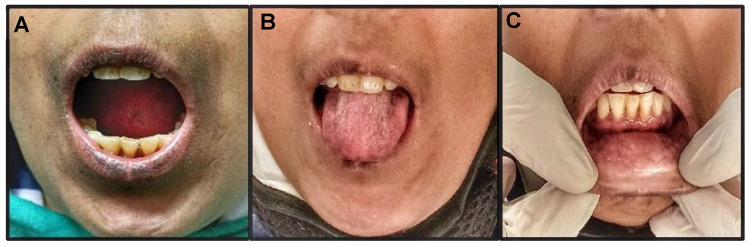
(**A**)—Reduced mouth opening, (**B**)—depapillated tongue with restricted protrusion, (**C**)—mild gingival recession and fibrotic labial mucosa.

**Figure 3 ijerph-19-10066-f003:**
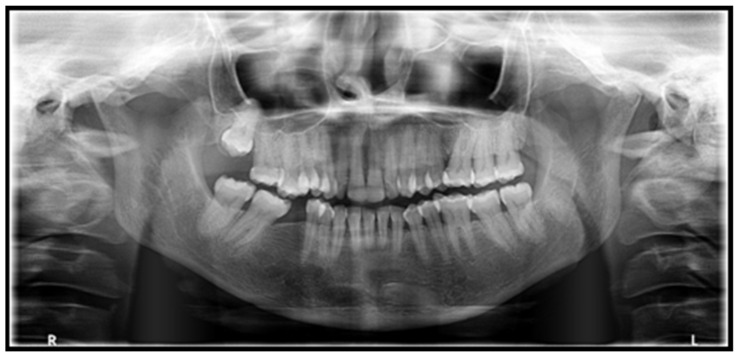
OPG reveals generalized PDL space widening. Mild flattening of the left condyle was also appreciated.

**Figure 4 ijerph-19-10066-f004:**
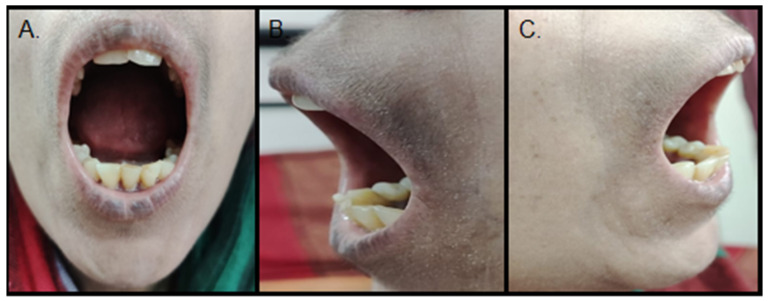
(**A**)—Frontal view. (**B**,**C**)—profile views showing improved mouth opening (30 mm) post muscle-stretching exercises and use of toothpaste and powered toothbrush.

**Table 1 ijerph-19-10066-t001:** Summary of investigations and treatment regimen.

S. No.	Investigation	Results
**1.**	**HIV Status**	
	HIV Serology	HIV-1 + ve (retro + ve since 2012) TLE regimen of ART
	CD4 cell count	440 cells/mm^3^
	HIV viral load	2,214,173 copies/mL
**2.**	**Complete Blood Count (CBC)**	
	Hemoglobin (Hb)	6.8 g/dL
	Total leucocyte count (TLC)	4490 cells/μL
	Platelets	1.3 lakh platelets/μL
	Erythrocyte sedimentation rate (ESR)	85 mm/hr
	Bleeding time (BT)	80 s
	Clotting time (CT)	42 s
	International normalized ratio (INR)	1.5
**3.**	**Liver function test (LFT)**	
	Albumin	2.3 g/dL
	Globulin	1.0 g/dL
	Bilirubin	2.6 mg/dL
	Alanine transaminase (ALT)	71 units/L serum
	Aspartate aminotransferase (AST)	33 units/L serum
	Alkaline phosphatase (ALP)	672 IU/L
	Hepatitis B surface antigen (HBs Ag)	−ve
	Anti HCV antibody	−ve
	α-Feto protein (AFP)	34 ng/mL
**4.**	**Kidney function test (KFT)**	
	Na^+^/K^+^	141/4.1 mEq/L
	Urea	21 mg/dL
	Creatinine	0.4 mg/dL
	Urine (R/M)	Normal range
**5.**	**Immunologic profile**	
	Antinuclear Antibodies (ANA)	1:1000 (Intensity 4+)
	Extractable Nuclear Antigens (ENA) profile (ELISA)	
	Anti ds-DNA autoantibody	−ve
	Ro (SSA) and La (SSB) autoantibody	−ve
	Anti Smantibodies	−ve
	Auto-RNP Ab	−ve
	Anti Scl-70- Ab	+ve (32.96 units/mL)
	Anti-mitrochondrial Ab (AMA)	+ve (suggestive of autoimmune hepatitis)
	Anti smooth muscle Ab (ASMA)	+ve (suggestive of autoimmune hepatitis)
	Anti cyclic citrullinated peptide (Anti-CCP)	−ve
	Anti cerulopasmin Ab	−ve
**6.**	**Radiographic investigations**	
	USG Abdomen	Mild coarse liver echotexture with surface nodularity, mild ascites
	Magnetic resonance cholangiopancreatography (MRCP)	Gall bladder cholestasis; hepatomegaly with chronic liver disease; splenomegaly with pulmonary hypertension; bilateral single renal cortical cyst; normal common bile duct (CBD); right and left hepatic ducts and intrahepatic bile duct
	High-resolution computed tomography (HRCT)	Ground glass opacity with peripheral and sub-pleural distribution suggestive of early stage interstitial lung disease
**7.**	**Liver Biopsy**	Revealed liver cirrhosis with activity compatible with autoimmune hepatitis
**8.**	**Current Treatment Regimen**	
	Table Methotrexate	15 mg weekly
	Table Folvite	5 mg weekly
	Table Omnacortical	5 mg OD
	Table Hydroquinone (HCQ)	200 mg OD HS
	Table Calcitin-D	500 mg BD
	Table Autrin	10 mg OD
	Table Etoricoxib	90 mg OD HS
	Table Pantop	40 mg OD

**Table 2 ijerph-19-10066-t002:** Reported studies and cases of HIV infection and SSc.

S. No	Author(s) and Year	Age/Sex	HIV Status	Systemic Sclerosis Features	Other Associated Ailments
1.	Sikdar et al., 2005 [18] Simultaneous diagnosis of HIV and SSc	45/F	CD4+ lymphocyte count increase observed after 6 months HAART.	The patient developed symptoms of SSc in the background of immune suppression and responded well to steroids and HAART therapy	-
2.	Mosquera JA, et al., 2010 [17]	44/M	HIV +ve, CD4 cell count = 934 cells/mm^3^, viral load < 40 copies/mL	Tumification of forearms, thighs, and legs; Raynaud’s phenomena +ve; skin biopsy revealed scleroderma features	+ve HCV serology
3.	Mosquera JA, et al., 2010 [17]	42/M	Stage II HIV infection	Skin thickening with spasticity in legs; skin biopsy revealed SSc features	+ve HCV and HBV serology; type II diabetes mellitus
4.	Okongo LO, et al., 2014 [5]	9/F	Perinatally acquired HIV; on ART therapy; CD4 cell count = 879 cells/mm^3^; lower than detectable viral load	Raynaud’s phenomena +ve with fingertip ulceration and digital ischemia; thickened extremities and torso skin; B/L sclerodactyly with limited extension and flexion of fingers	Completed TB therapy at 6 months
5.	Dembelae IA et al., 2018 [16] Concomitant diagnosis of HIV and systemic sclerosis.	56/F	HIV stage III	Sclerodactyly; +ve Raynaud’s phenomena; +ve anti Scl-70 Abs	+ve HBV serology
6.	Yao Q, et al., 2008 [24] Retrospective study; out of 888 HIV diagnosed cases, only 1 case of SSc was seen	45/F	HIV +ve (acquired from heterosexual partner)	Progressive stiffening of face and extremities skin, mask-like face	Renal insufficiency
7.	Yen et al., 2016 [25] Only 4 cases of SSc out of 20,444 HIV cases	3 males and 1 female	1 patient on HAART and 3 patients without HAART	-	-

**Table 3 ijerph-19-10066-t003:** A summary of the sequence congruity between viral proteins and human autoantigen epitopes.

Author (s) and Year	Autoantigen	Viral Antigen
Douvas et al., 1996 [28]	U1 RNP 70 kD; seen in MCTD and SSc	HIV gp120/41 envelope complex
Query CC et al., 1987 [29]	U1 RNP 70 kD; seen in MCTD and SSc	Retroviral p30 gag protein
Maul GG et al., 1989 [30]	DNA Topoisomerase I 110 kD; seen in diffuse SSc	Retroviral p30 gag protein

## Data Availability

Not applicable.

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
