# Peer review of "HIV-Associated Systemic Sclerosis: Literature Review and a Rare Case Report"

_ijerph, 2022, doi:10.3390/ijerph191610066_

Round 1
Reviewer 1 Report
Dear Authors,
I read your manuscript titled "HIV-Associated Systemic Sclerosis: Report of Rare Case with A Brief Literature Review." You presented this case very well. This manuscript emphasises the role of dental medicine professionals in recognising the very first signs of HIV-associated disorders that manifiset in the oral cavity.
Author Response
I read your manuscript titled "HIV-Associated Systemic Sclerosis: Report of Rare Case with A Brief Literature Review." You presented this case very well. This manuscript emphasizes the role of dental medicine professionals in recognizing the very first signs of HIV-associated disorders that manifest in the oral cavity.
Respected Reviewer,
Thanks for your kind recognition

Reviewer 2 Report
Case reports of interest to the dental world, where these correlations are often overlooked.Changes need to be made
Adding specific keywords there are few
Introduction: complications related to the periodontal and hard tissues to be added. Incidence?
Case report: how was it treated professionally? the maintenance plan? Toothpaste combined with electric toothbrush? why limit ourselves only to the opening? why not consider the integration of probiotics for the maintenance of eubosis?
Discussion, to be added as future objectives, the proactive action for the maintenance of eubiosis through paraprobiotics and postbiotics already studied by the group of Prof. Scribante et all.
Conclusions. Add proactive action for maintenance bibliography, to add references
Add single images in high definition
Author Response
Case reports of interest to the dental world, where these correlations are often overlooked.Changes need to be made
Adding specific keywords there are few-
Specific keywords are added.
Introduction: complications related to the periodontal and hard tissues to be added. Incidence?
Incidence and complications related to the periodontal and hard tissues added in the introduction..
Case report: how was it treated professionally? the maintenance plan? Toothpaste combined with electric toothbrush? why limit ourselves only to the opening? why not consider the integration of probiotics for the maintenance of eubosis?
Probiotics were prescribed as a maintenance therapy for the regulation of eubiosis. Oral hygiene maintenance by toothpaste combined with a powered toothbrush was advised to the patient.
Discussion, to be added as future objectives, the proactive action for the maintenance of eubiosis through paraprobiotics and postbiotics already studied by the group of Prof. Scribante et al.
Role of paraprobiotics and postbiotics as a maintenance therapy for the regulation of eubiosis incorporated in the treatment part (in discussion)
Conclusions. Add proactive action for maintenance bibliography, to add references
Recent references incorporated.
Add single images in high definition
Better quality images added

Reviewer 3 Report
This case report presents
The title can be presented in a better way by “HIV-Associated Systemic Sclerosis: Report of Rare Case with A 2 Brief Literature Review” as “HIV-Associated Systemic Sclerosis: Literature review and A Rare Case Report”.
Please add the reference to the Line “It may diminish the autoimmune episode by restoring the dysregulated immune function”.
The introduction is shoter. More related literature can be added on HIV-Associated Systemic Sclerosis general symptoms and treatment options, effectiveness, prognosis, etc.
Is it possible to add the references on the treatment regimes?
Figures. Please add better images. Figure 1D is stretched sideways.
Intraoral images are not clear (Figure 2-3).
Discussion.
Line 186-195 can be moved in the Introduction.
It will be helpful to discuss the recent and future general treatments for HIV and treatments for sclerosis?
Author Response
The title can be presented in a better way by “HIV-Associated Systemic Sclerosis: Report of Rare Case with A 2 Brief Literature Review” as “HIV-Associated Systemic Sclerosis: Literature review and A Rare Case Report”.
The suggested title has been incorporated
Please add the reference to the Line “It may diminish the autoimmune episode by restoring the dysregulated immune function”.
Reference added
The introduction is shorter. More related literature can be added on HIV-Associated Systemic Sclerosis general symptoms and treatment options, effectiveness, prognosis, etc.
An elaborate introduction is added
Figures. Please add better images. Figure 1D is stretched sideways.
Intraoral images are not clear (Figure 2-3).
Better images are added
Discussion.
Line 186-195 can be moved in the Introduction.
Introduction is made elaborate, however, these lines are in the discussion section.
It will be helpful to discuss the recent and future general treatments for HIV and treatments for sclerosis?
Recent and future treatment for HIV mentioned, and treatment for sclerosis is also incorporated in the discussion.

Reviewer 4 Report
The topic of the manuscript is a rare case of a female patient with HIV-associated systemic sclerosis, who was referred to the authors for pain management in her mandibular teeth.
The title and the abstract of the article are informative. The Introduction briefly presents the issue of the relationship between rheumatic symptoms of autoimmune diseases and HIV infection/highly active antiretroviral therapy. The section "Case report" precisely describes the mentioned patient, including extraoral, intraoral and radiological examination pictures, but the presentation of this section should be improved. The Discussion is interestingly written, however, it should be supplemented with more recent references. The Conclusions seem to be the "take-home" messages.
Some following points must be clarified/corrected for the further processing of this article.
Merits-related comments:
- Please complete keywords with the proper MeSH terms, necessary for indexing in the databases.
- Please indicate in which databases were searched for case reports with patients with HIV-associated systemic sclerosis (in the Introduction).
- Table 1 requires graphical editing and correction, e. g. missing dose for Tab. Hydroquinone.
- Legends under Figures need to be sorted and specified, e. g. successive images in Figure 2 should have a corresponding description under letters (a), (b) and (c).
- It is suggested to add more recent articles from 2019-2022 to the references in the Introduction and the Discussion.
- At the end of the Discussion, the potential difficulties in medical and dental care should be explained more clearly, how they can be addressed in the multidisciplinary approach.
Technical comments:
- Affiliates of the authors are duplicated and require a connection to one.
- The abstract should be a single paragraph and should follow the style of structured abstracts but without headings.
- The manuscript requires careful editorial correction – numerous repetitions (e. g. "of" in line 50), double spaces (e. g. lines 40, 42, …), incomplete quotes (line 177) or typos (e. g. …). Also, unnecessary italics for symptom designations.
- All acronyms or abbreviations (e. g. HIV, AIDS) should be defined the first time they appear in each of three sections: the abstract; the main text; the first figure or table. When defined for the first time, they should be added in parentheses after the written-out form.
- In the text, reference numbers should be placed in square brackets [ ], and placed before the punctuation; for example [1], [1–3] or [1,3] – e. g. in line 115 "(2005)" to remove. However, references to figures and tables should be in round brackets ( ) not in square ones.
- In Table 2 no consequence for authors' citations – everywhere, there should be a first author plus "et al.".
- The citation list must be corrected. References should be described as follows:
1. Author 1, A.B.; Author 2, C.D. Title of the article. Abbreviated Journal Name Year, Volume, page range. - In Author Contributions, the following statements should be used "Conceptualization, X.X. and Y.Y.; Methodology, X.X.; Software, X.X.; Validation, X.X., Y.Y. and Z.Z.; Formal Analysis, X.X.; Investigation, X.X.; Resources, X.X.; Data Curation, X.X.; Writing – Original Draft Preparation, X.X.; Writing – Review & Editing, X.X.; Visualization, X.X.; Supervision, X.X.; Project Administration, X.X.; Funding Acquisition, Y.Y.".
Author Response
The topic of the manuscript is a rare case of a female patient with HIV-associated systemic sclerosis, who was referred to the authors for pain management in her mandibular teeth.
The title and the abstract of the article are informative. The Introduction briefly presents the issue of the relationship between rheumatic symptoms of autoimmune diseases and HIV infection/highly active antiretroviral therapy.
The section "Case report" precisely describes the mentioned patient, including extraoral, intraoral and radiological examination pictures, but the presentation of this section should be improved.
Case reported presentation is improved.
The Discussion is interestingly written, however, it should be supplemented with more recent references. The Conclusions seem to be the "take-home" messages.
Recent references are added in the discussion.
Some following points must be clarified/corrected for the further processing of this article.
Merits-related comments:
- Please complete keywords with the proper MeSH terms, necessary for indexing in the databases.
Complete keywords with the proper MeSH terms included in the introduction.
- Please indicate in which databases were searched for case reports with patients with HIV-associated systemic sclerosis (in the Introduction).
Google scholar and PubMed search engines were used, included in the introduction.
- Table 1 requires graphical editing and correction, e. g. missing dose for Tab. Hydroquinone.
Graphical editing and correction done in Table 1. Missing dose of Hydroxyquinone added
- Legends under Figures need to be sorted and specified, e. g. successive images in Figure 2 should have a corresponding description under letters (a), (b) and (c).
Legends under figures sorted and specified.
- It is suggested to add more recent articles from 2019-2022 to the references in the Introduction and the Discussion.
Recent references added in both introduction and Discussion.
- At the end of the Discussion, the potential difficulties in medical and dental care should be explained more clearly, how they can be addressed in the multidisciplinary approach.
Potential difficulties in medical and dental care of systemic sclerosis patients and their management included in the discussion.
Technical comments:
- Affiliates of the authors are duplicated and require a connection to one.
incorporated
- The abstract should be a single paragraph and should follow the style of structured abstracts but without headings.
Suggestion incorporated
- The manuscript requires careful editorial correction – numerous repetitions (e. g. "of" in line 50), double spaces (e. g. lines 40, 42, …), incomplete quotes (line 177) or typos (e. g. …). Also, unnecessary italics for symptom designations.
The manuscript has been run on Grammarly and all possible corrections included.
- All acronyms or abbreviations (e. g. HIV, AIDS) should be defined the first time they appear in each of three sections: the abstract; the main text; the first figure or table. When defined for the first time, they should be added in parentheses after the written-out form.
Suggestion incorporated
- In the text, reference numbers should be placed in square brackets [ ], and placed before the punctuation; for example [1], [1–3] or [1,3] – e. g. in line 115 "(2005)" to remove. However, references to figures and tables should be in round brackets ( ) not in square ones.
Suggestion incorporated
- In Table 2 no consequence for authors' citations – everywhere, there should be a first author plus "et al.".
Suggestion incorporated
- The citation list must be corrected. References should be described as follows:
Author 1, A.B.; Author 2, C.D. Title of the article. Abbreviated Journal NameYear, Volume, page range.
References checked
- In Author Contributions, the following statements should be used "Conceptualization, X.X. and Y.Y.; Methodology, X.X.; Software, X.X.; Validation, X.X., Y.Y. and Z.Z.; Formal Analysis, X.X.; Investigation, X.X.; Resources, X.X.; Data Curation, X.X.; Writing – Original Draft Preparation, X.X.; Writing – Review & Editing, X.X.; Visualization, X.X.; Supervision, X.X.; Project Administration, X.X.; Funding Acquisition, Y.Y.".
Author contribution included as per the suggestion.

Reviewer 5 Report
This is an interesting case report of HIV-associated systemic sclerosis. However, the presentation of the clinical photographs should improve. Moreover, the discussion has too many paragraphs that preclude consistency in interpreting and discussing the findings. thus, the discussion should be reorganised to include focused paragraphs rather than short multiple parts.
Other points:
L 50: it would be useful for readers to cite these 4 cases.
L53: report the date of referral.
L98: when was the follow-up visit scheduled?
L127-132: this is not a proper way for the discussion. Rephrase and elaborate, please.
Periodontal ligament space widening and mandibular erosions are the distinctive radiographic manifestations of systemic sclerosis [22]. 202 Periodontal ligament space widening is the most frequently occurring radiographic 203 presentation, and systemic sclerosis should be given a place in the differential diagnosis 204 where the radiographic features demonstrate widened PDL space with intact lamina dura 205 (particularly in non-mobile posterior teeth with widened PDL space in more than one 206 segment) [26].
These sentences should be included in the same paragraph.
Figure 1 and Figure 2: Hand and intra-oral photos are poor. The clinical photos should be displayed in better quality.
Author Response
This is an interesting case report of HIV-associated systemic sclerosis. However, the presentation of the clinical photographs should improve. Moreover, the discussion has too many paragraphs that preclude consistency in interpreting and discussing the findings. thus, the discussion should be reorganised to include focused paragraphs rather than short multiple parts.
Better quality images included. Discussion reorganized including focused paragraphs.
Other points:
L 50: it would be useful for readers to cite these 4 cases.
The 4 case reports are cited.
L53: report the date of referral.
Referral date included.
L98: when was the follow-up visit scheduled?
Follow up visit included in the case report.
L127-132: this is not a proper way for the discussion. Rephrase and elaborate, please.
Suggestion incorporated.
Periodontal ligament space widening and mandibular erosions are the distinctive radiographic manifestations of systemic sclerosis [22]. 202 Periodontal ligament space widening is the most frequently occurring radiographic 203 presentation, and systemic sclerosis should be given a place in the differential diagnosis 204 where the radiographic features demonstrate widened PDL space with intact lamina dura 205 (particularly in non-mobile posterior teeth with widened PDL space in more than one 206 segment) [26].
These sentences should be included in the same paragraph.
These sentences are included in one paragraph.
Figure 1 and Figure 2: Hand and intra-oral photos are poor. The clinical photos should be displayed in better quality.
Better quality images are included.

Round 2
Reviewer 2 Report
The manuscript has been correctly revised according to the comments suggested, it can be processed for publication
Author Response
The manuscript has been correctly revised according to the comments suggested, it can be processed for publication.
Respected Reviewer
Thank you.

Reviewer 4 Report
The Authors have greatly improved the manuscript by referring to most of the comments made during the review. There are still minor flaws, such as incorrect reference list formatting or a footnote in Tables 2 and 3 before the year (instead of after). Also, Figure 1 hides its description in the footer.
Author Response
The Authors have greatly improved the manuscript by referring to most of the comments made during the review. There are still minor flaws, such as incorrect reference list formatting or a footnote in Tables 2 and 3 before the year (instead of after). Also, Figure 1 hides its description in the footer.
Respected Reviewer
Thank you for the acknowledgement. The suggested minor changes have also been incorporated,

Reviewer 5 Report
The authors have addressed my comments, and the the manuscript has significantly improved. I only have one minor comment, Figure 1 is imposed over the footnotes text.
Author Response
The authors have addressed my comments, and the the manuscript has significantly improved. I only have one minor comment, Figure 1 is imposed over the footnotes text.
Respected Reviewer
Thank you for the acknowledgement. However, the minor comment raised has also been rectified.
Thank you

This manuscript is a resubmission of an earlier submission. The following is a list of the peer review reports and author responses from that submission.